# Vectors of Dutch Elm Disease in Northern Europe

**DOI:** 10.3390/insects12050393

**Published:** 2021-04-29

**Authors:** Liina Jürisoo, Ilmar Süda, Ahto Agan, Rein Drenkhan

**Affiliations:** 1Institute of Forestry and Rural Engineering, Estonian University of Life Sciences, Fr.R. Kreutzwaldi 5, 51006 Tartu, Estonia; ahtoagan@hotmail.com (A.A.); rein.drenkhan@emu.ee (R.D.); 2Ilmar Süda FIE, Rõõmu tee 12-5, 50705 Tartu, Estonia; ilmar.suda@eesti.ee

**Keywords:** *Scolytus* spp., DED, *Xyleborinus saxesenii*, *Xyleborus dispar*, pheromone trap, *Ophiostoma novo-ulmi*, climate change, PacBio sequencing

## Abstract

**Simple Summary:**

Dutch elm disease (DED) has been killing elms for more than a century in northern Europe; the trees*’* health status has worsened substantially in recent decades. Elm bark beetles *Scolytus* spp. are vectors of DED. Our aim was to estimate the distribution range of elm bark beetles and to detect potential new vectors of DED agents in northern Europe. Beetles were caught with bottle traps and manually. Then DNA from each specimen was extracted and analysed by the third generation sequencing method. DED agents were detected on the following bark beetles for Europe: *Scolytus scolytus*, S*. triarmatus*, *S. multistriatus*, *S. laevis*, and on new vectors: *Xyleborus dispar* and *Xyleborinus saxesenii*. The spread of *Scolytus triarmatus*, *S. multistriatus* and *Xyleborinus saxesenii* has been remarkable for the last two decades, and *S. triarmatus* and *X. saxesenii* are relatively recent newcomers in the northern Baltics. The problem is that the more vectoring beetles there are, the faster spread of Dutch elm disease from tree to tree.

**Abstract:**

Potential Dutch elm disease vector beetle species were caught with pheromone bottle traps and handpicked in 2019: in total, seven species and 261 specimens were collected. The most common was *Scolytus triarmatus*, but by percent, the incidence of *Ophiostoma novo-ulmi* was highest in *Scolytus scolytus*, followed by *Xyleborinus saxesenii* and *S. triarmatus*. We analysed the beetles*’* DNA using PacBio sequencing to determine vector beetles of *Ophiostoma novo-ulmi*. *Ophiostoma novo-ulmi* was found on six out of seven analysed beetle species: *Scolytus scolytus*, S*. triarmatus*, *S. multistriatus*, *S. laevis*, *Xyleborinus saxesenii* and *Xyleborus dispar*. The last two beetles were detected as vectors for *Ophiostoma novo-ulmi* for the first time. Previous knowledge on the spread of beetles is discussed.

## 1. Introduction

There are three native *Ulmus* species (*U. glabra* Huds., *U. laevis* Pall. and *U. minor* Mill.) in northern Europe. *Ulmus glabra,* having more northern range, has spread throughout Estonia; *U. laevis,* having more southern range is growing mainly along riversides; *U. minor* native range reaching at its northernmost extent the Baltic Sea and not to northern Baltics [1,2]. *Ulmus* expanded to Estonia during the Pre-Boreal period, spread rather rapidly, and elms started to decline at the end of the Atlantic period [3].

Elms (*Ulmus* spp.) as a keystone native forest or amenity species have been under attack globally for more than a century due to the Dutch elm disease causal agent *O. ulmi s.l.* [4,5,6,7,8]. Dutch elm disease is a lethal vascular wilt disease, the first symptoms of which are yellowing and browning leaves, a cross-section of an elm twig showing brown spots or streaks in the recent wood rings [9,10].

The first pandemic caused by *O. ulmi* killed 10–40% of elms by 1960 [8,11,12]; the second more severe pandemic killed some 80–90% of mature elms by the beginning of the 21st century in the UK [13,14], as well as hundreds of millions in North America [15]. All native elm species (*U. glabra*, *U. laevis*, *U. minor*) in Sweden endured enormous decimation due to DED and were registered on the Red List since 2010 as no long-term viable species [16]. DED has been devastating to various *Ulmus* species in Estonia, other Baltic countries and in western Russia since the last decade of the 20th century [17,18,19,20,21]. Ophiostomatoid fungal communities (incl. DED) depend on host trees and vector beetles. Coexistence of fungi and their vectors has been studied [22,23] on conifers [24,25] and considerably less on hardwood species [26].

The mycobiota of xylomycetophagous bark beetles is well studied on ambrosia beetles [27]; less studied in Europe, in particular, are phloem-breeding bark and woodboring beetle-associated fungi; those studies were mainly based on morphological criteria [26]. Bark beetles (Coleoptera: Curculionidae, Scolytinae) are distributed worldwide and form many cosmopolitan genera [28]. Although there are many species of scolytids associated with angiosperm trees (hardwoods) [29], nearly all of them are secondary bark beetles as they colonise stressed or damaged trees [6]. A great majority of scolytid beetles on hardwoods are minor pests and are of no economic importance; an exception—some bark beetles connected with elms [5,29]. Elm bark beetles known to vector DED agents include *Scolytus kirchi* Stalitzky, *S. laevis* Chaupis, *S. multistriatus* Marsham with its variety *triornatus* Eichhoff, *S. pygmaeus* Fabricius, *S. schevyrewi* Semenov, *S. scolytus* Fabricius, *S. triarmatus* Eggers, *S. ulmi* L. Redtenbacher and *Hylurgopinus rufipes* Eichhoff [9,30,31,32,33,34]. Only a few of those are significant [30,35,36], the most important being *Scolytus multistriatus* [4,32,37], *S. scolytus* [30,38] and *S. pygmaeus* [4,36].

Several other species of bark beetles which carry fungal associates in mycangium [39] feed in phloem tissues (inner bark), though some larvae engrave outer sapwood [40], being phloeophagous, preferring mostly species from Ulmaceae [5,9]. Elm bark beetles carry DED pathogen spores on the surface of their body and in their gut [30,36,41,42] from diseased to healthy trees, feeding on and tunnelling in twig crotches of healthy elms, transferring fungal spores to xylem tissues [43]. Fungi spreads inside a branch causing blockage of the conducting system because of the formation of tyloses that cause leaves to wilt and die [32].

Some beetle species that live in association with fungi, e.g., ambrosia beetles are xylomycetophagy—feeding on mycelia, consuming wood incidentally [40]. These are polyphagous species of beetles that inhabit many deciduous tree species, incl. Ulmaceae; those common to northern Europe are *Xyleborus dispar* Fabricius, *Xyleborinus saxesenii* Ratzeburg and *Trypodendron signatum* Fabricius [44,45,46]. There is a slight possibility that DED could be spread by enthomophagous species that follow bark beetles in their tunnels, e.g., *Salpingus planirostris* and *S. ruficollis*, which are common on deciduous trees in northern Europe. Both had been caught in tunnels of *Scolytus scolytus* in the late stages of development [47]. Similarly, there is a minor possibility that *Paromalus parallelepipedus* and *Hololepta plana* feeding rarely under the bark of elms [48,49] are potential predatory species on *Scolytus* species. Together with elm bark beetles, there are at least seven species of bark beetles in the northern Baltics that can potentially spread the Dutch elm disease agent.

Climate change has caused relatively warmer winters and springs than summers and falls, increasing mean annual temperatures [50,51,52]. This has an impact on trees in forest ecosystems and in urban areas [53]. For example, *Ulmus glabra* has already started to extend the northern range of its distribution [54,55]. At the same time, climate extremes like unusual fluctuation of temperatures, heavy rains [56], and severe storm events [51] put the hosts under stress and make them susceptible to pests and diseases [51,57]. Those extremities have already caused pathological consequences in Estonian forests [58,59,60,61,62,63]; new diseases or pests may affect elms, and some may become more aggressive [64,65].

Climate also influences the beetles*’* outbreaks, their aggressiveness, population dynamics and migration [51] that become more frequent [66]. Usually, the number of scolytid species increased from north to south [67], depending on suitable number of tree species [28]. Warmer climate will probably extend the northern range of *Scolytus* spp. and affect elm trees in the cooler parts of northern Europe [68], as has happened in northwest Russia [52]. As *Scolytus scolytus*, *S. laevis*, *S. multistriatus* are temperature dependent, beetles may be active for a longer period of time than previously, as they start to fly when the mean temperature is at or above 16 °C [32,33,47] and during an extended warmer period due to climate change.

The aim of the paper is to estimate the distribution of elm bark beetles in Estonia and to detect other potential beetle vectors of DED agents in northern Baltics and the north-western part of Russia.

## 2. Materials and Methods

### 2.1. Study Sites and Sampling

The study sites (Figure 1) were chosen among the locations assessed for DED during 2013–2018 [20,21]. All the sampling sites were located in either urban or rural parks; DED had been found in eight park sites and two well-known elm sites were without previous disease conformation. For trapping beetles, species-specific pheromones (semiochemicals) were used [69].

Thirty-nine bottle traps (see Appendix A) that contained a 1.5 L plastic bottle with a cut window, plus a smaller bottle with 96% ethanol on the bottom and a pheromone with attractants for *Scolytus* spp. were hung on trees in 10 sites at least 50 m from each other at 3 m above ground as the most effective height [70,71]. Lures consisted of two semipermeable plastic pouches containing a mixture of cubeb oil, 1-hexanol, multistriatin and 4-methyl-3-heptanol (Synergy Semiochemicals Corp., Burnaby, BC, Canada). The lures attract beetles which, while flying towards the lure, hit the plastic sheet, fall into the container with 96% ethanol and sink. We assume that trapped beetles*’* cross contamination is eliminated in ethanol. Beetles died in ethanol quickly, traps were checked systematically, and the number of beetles was quite low (generally 1–2 individuals and same species per trap). Twenty-three traps were placed in Tallinn, northern Estonia; 16 traps, throughout Estonia (Figure 1).

Traps were hung in the beginning of June 2019 on what were at the time visually healthy elm trees; sampling was carried out from the second half of June until the beginning of September, at least every three weeks, five times total during the season.

*Scolytus*-like beetles were immersed in 96% ethanol for sterilisation and were put in separate tubes and stored in −20 °C until further processing.

To secure a sufficiently large sample size for DNA analysis, we also hand-picked beetles from bark surface of trees at three sites in Estonia (four trees) and one site in St. Petersburg, Russia (three trees). Each specimen was captured into a separate sterile tube.

### 2.2. Beetles’ Identification

All beetles caught were identified using an Olympus stereo zoom microscope SZ60 (Olympus Corporation, Japan) with 100× maximum magnification, based on the following identification keys [72,73,74,75,76]. If necessary, genitals were separated, and the sex of the bark beetles were determined. There were 319 determined beetles in total.

### 2.3. Molecular Analyses

DNA was extracted separately from each beetle*’*s whole body [77] using GeneJET Genomic DNA purification kit (Thermo Fischer Scientific, Vilnius, Lithuania). All 259 collected and determined possible vector beetles were DNA extracted from which 109 were randomly selected, covering different beetle species, sex, locations, and sampling for future analyses.

Primers ITS4ngsUni [78] and ITS1catta [79] were used to amplify fungal DNA and the PCR products were sequenced using PacBio sequencing in the University of Oslo in Norway. Both reverse and forward primers were equipped with 109 different MID tags with 10–12 base length (different pair per sample) that had at least 4 base differences from one another. PacBio has recently been successfully used in metabarcoding analysis of microorganisms on various trees and plants, as the long DNA barcodes of 500–1500 bp can improve OTU identification on a species level [80,81,82].

Conventional PCR was carried out according to [82] with two replicates for each sample in 25 µL reaction volume containing 0.5 µL of forward and reverse primer and 5 µL of HOT FIRE Pol Blend Master Mix Ready to Load (Solis BioDyne, Tartu, Estonia). Amplification was performed as follows: 15 min at 95 °C, followed by 25 cycles of 30 s at 95 °C; 30 s at 55 °C; 1 min at 72 °C, and a final step at 72 °C for 10 min. Positive and negative controls were used throughout the analysis to exclude possible tag switches and sample contamination during the PCR process.

The PCR reactions were checked for the presence of a product on 1% agarose gel. In the case of no visible band, we repeated the amplification by increasing the number of cycles up to 35. The PCR products were purified using FavorPrep™ GEL/PCR Purification Kit (Favorgen, Vienna, Austria) following the manufacturer*’*s instructions.

The amplicons were pooled into one sequencing library. Library preparation followed the protocols established for the RSII instrument of PacBio third-generation sequencing platform (Pacific Biosciences, Inc. Menlo Park, CA, USA). The diffusion method was used in loading the library to SMRT cells. Sequencing was performed using P6-C4 chemistry for 10 h following Tedersoo et al. [83].

### 2.4. Bioinformatics and Statistical Analysis

Bioinformatics was carried out by using various programs implemented in Pipecraft v1.0 [84].

Using mothur (v1.36.1) [85], reads < 100 bp were removed and longer sequences were demultiplexed allowing 2-base differences to index and 3-base differences to primer. UCHIME [86] was used in de novo chimera filtering. The full-length Internal Transcribed Spacer (ITS) region was extracted from the rRNA genes with program ITSx (v1.0.11) [87]. CD-HIT (v4.6) [87] was used to cluster sequences into Operational Taxonomic Units (OTUs) based on 97% sequence similarity. OTUs were taxonomically identified based on representative sequences against the UNITE v.7 database [88]. OTUs were considered as members of fungi if their representative sequences matched the best fungal taxa at e-value < e−50. Representative sequences that had >97% sequence similarity to reference sequences were assigned to species hypotheses (SHs) based on UNITE [89]. Higher level classification of fungi was based on the e-value and sequence similarity criteria of Tedersoo et al. [78]. Differences of percentage of *O. novo-ulmi* between sampling methods, sampling areas, beetle species and genders were analysed using ANOVA with Tukey HSD in Excel and were considered significant with *p* value ≤ 0.05.

## 3. Results

### 3.1. Collected Beetle Species

In total, 319 specimens of beetles (28 different species) were caught, from which 93 specimens of 23 beetle species were captured with pheromone-baited bottle traps (28 of the 39 traps contained beetles). The number of potential vector beetle individuals for DED was 261, from which 81% of beetles were handpicked in four sites from seven different trees; 9% were trapped. The number of potential DED vector beetles collected with traps and symptomatic trees are presented by species, country, and gender in Table 1. The other beetle species collected are listed in Appendix A.

### 3.2. Ophiostoma novo-ulmi on Vector Beetles

The number of sequenced individuals was 109; the selection was based on beetle specimens that covered different beetle species, sex, locations and different collecting methods.

The entire sequenced dataset consisted of 33,202 high quality sequences across 109 beetle specimen samples and 655 OTUs (see Genbank accession number PRJNA719602).

*Ophiostoma novo-ulmi* was found on six out of seven beetle species. Only *S. pygmaeus* had no *O. novo-ulmi*. Among the most caught species was *S. triarmatus;* the pathogen was detected in 77% of analysed specimens.

In total, 2757 *O. novo-ulmi* ITS sequences were found in the dataset, constituting 8.3% of all sequences within the dataset (see https://dx.doi.org/10.15156/BIO/807454, accessed on 15 April 2021). *Ophiostoma novo-ulmi* was the most prevalent species in this dataset. The highest average percentage of *O. novo-ulmi* per sample was found on beetle species *S. scolytus,* followed by *X. saxesenii* and *S. triarmatus* with 26.6%, 20.5% and 18.2%, respectively; although considerable, these differences among species were not statistically significant, possibly due to a large variation in sample sizes among species (F_5.101_ = 1.89; *p* = 0.101; Figure 2). The difference between genders was also not statistically significant (F_5.101_ = 0.001; *p* = 0.993). Comparison between handpicked and trapped specimens resulted no significant difference in percentages of *O. novo-ulmi* (F_1_._80_ = 0.04; *p* = 0.848). The beetles handpicked directly from trees had 8.7% of sequences identified as *O. novo-ulmi,* whereas 8.1% of *O. novo-ulmi* was found on beetles collected with traps. When comparing the relative abundance of *O. novo-ulmi* across 15 different sampling sites and all beetle species, the highest percentage of *O. novo-ulmi* was found in site Tallinn (Kopli), North Estonia, followed by site Vastseliina, Southeast Estonia and site St. Petersburg, Russia with 27.7%, 18.7% and 15.4%, respectively. According to ANOVA with Tukey HSD *O. novo-ulmi* percentage differences between sites were not statistically significant (F_8.91_ = 1.42; *p* = 0.196). For more precise distribution data of *O. novo-ulmi* across the sampling sites, beetle species and gender, see Appendix A.

## 4. Discussion

### 4.1. Beetles as Vectors of O. novo-ulmi and Pathogen Detection

When referring to beetles that spread Dutch elm disease, in particular, we mean those species that can inhabit living elms, their bark and/or wood.

According to PacBio sequencing, we found the pathogen in six commonly captured beetle species (*S. scolytus, S. laevis, S. multisriatus, S. triarmatus, X. saxesenii*, *X. dispar*; see Figure 2). Among those, *X. saxesenii* and *X. dispar* were found as new vectors for Dutch elm disease in northern Europe. *Ophiostoma novo-ulmi* was not detected on *S. pygmaeus* in this study.

Bark and ambrosia beetles are an optimal vehicle for the transport of different organisms, incl. fungi, from one host to another [40].

Until now, the main culprits in the spread of Dutch elm disease have been *Scolytus* spp. [4,32,37]. Knowing the biology of Scolytinae and their suitable host trees, the range of these potential vector species is somewhat wider. In addition to *Scolytus* sp., host species from family Ulmaceae are inhabited by at least the following species of beetles common in northern Europe: *Xyleborus dispar*, *Xyleborinus saxesenii* and *Trypodendron signatum* [44,45,46]. These are polyphagous pests that inhabit many deciduous tree species but may rarely occur in conifers as well. *Xyleborus dispar* can also attack healthy trees, especially when those are close to stressed hosts [89,90].

Most ambrosia beetles do not feed on wood, but *Xyleborinus saxesenii* is an exception. The larvae feed on the ectosymbiotic fungus growing in wood and the wood itself, classifying this species as xylomycetophagous, rather than just mycetophagous (feeding only on fungi) [91]. *Xyleborinus saxesenii* is strongly attracted to ethanol-based baits and often accumulates in ethanol-baited traps in numbers greater than other ambrosia beetles [92]. Most cases where the species is reported as aggressive to stressed hosts include elms [93].

In Poland *O. novo-ulmi* subsp. *novo-ulmi* was isolated not from only the elm-infecting beetles but also from *Hylesinus crenatus* on *Fraxinus excelsior* and *Scolytus intricatus* on *Quercus robur* [22]. *Ophiostoma novo-ulmi* was found on an unknown vector and on other host species than elms, thus the pathogen occurrence in forest ecosystems is much broader than previously thought [22,94].

The pathogen detection from biological samples incl. vector beetles is highly crucial to estimate the disease spread and occurrence. Thus, we note that primers ITS1catta and ITS4ngsUni and PacBio sequencing platform worked well identifying *O. novo-ulmi* from different bark beetle species. The primers were able to distinguish *O. novo-ulmi* across 33,202 ITS sequences and 655 OTU-s. *Ophiostoma novo-ulmi* was the most prevalent species in this dataset: 8.3% of sequences were identified as *O. novo-ulmi* with the average length of *O. novo-ulmi* amplicons being 612 bp. We sequenced the ITS region, which does not differentiate *O. novo-ulmi* subspecies, but we have evidence that the subspecies may have different aggressiveness (see [17,95]). Thus, species-specific DNA primers are needed to differentiate *O. novo-ulmi* subspecies from biological samples for faster detection of the spread of pathogen.

### 4.2. Spread of Vectors for DED in Estonia and Northwest Russia

*Scolytus* bark beetles are the main vectors for the transmission of DED, introducing the pathogen into visually healthy trees during adult feeding and breeding.

*Scolytus multistriatus* was found to vector *O. novo-ulmi* in the current work and the pathogen was detected in about 8% of analysed individuals. The northernmost European finding of *Scolytus multistriatus* was recorded in a park in St. Petersburg in 1997 [96,97], and later in Vyborg (B.G. Popovichev, pers. comm.). It was first registered in Estonia, close to Tartu in about 1900 [98], then was rediscovered in southern Estonia: Taheva (1967) and Karisöödi (1996) [99]. Since 2012, the species has been relatively abundant on the Koiva wooded meadow near Vaitka, South Estonia. According to the latest data, in 2019, *S. multistriatus* has already reached central Estonia.

*Scolytus laevis,* of which 2.5% of the analysed individuals contained the pathogen, is the most widely spread species of its genera in Estonia and Russia [100] but has been found much farther north in Sweden and Norway than in the central parts of the Leningrad Region [96,97]. It was recorded for the first time in South Estonia at Heimtali, Viljandi County in 1936 [101], a few years later, in 1938, also from Viljandi, central Estonia [102]. *Scolytus laevis* was also found in northern Estonia [103] and is now widespread presumably throughout the Estonian mainland. Years ago, DED was rarely spread by this species [100] but it has been currently proved as a vector of DED in our region.

*Scolytus scolytus,* of which more than fourth of analysed specimens contained *O. novo-ulmi*. The northernmost recordings of *S. scolytus* in Europe were found in St. Petersburg*’*s city parks in 2000 [97,104] and later in Vyborg (B.G. Popovichev, pers. comm.). It should be mentioned that in Sweden, *S. scolytus* has been completely replaced by the related species *S. triarmatus*, which was reported from even farther northern Sweden, compared with *S. scolytus* in Estonia [104]. Only two dead specimens were found in Estonia on the island of Abruka in western Estonia (leg. et det. I. Süda): 1 ♀ under the bark of on old dead elm in 1994 and 1 ♂ under the bark of a thick branch of an old elm in 1997 [99,105,106].

*Scolytus triarmatus,* of which 18% of analysed specimens carried *O. novo-ulmi.* The first finding of *S. triarmatus* in Estonia (also in the Baltics) originated from Soontaga, southern Estonia in 2004 [106]. It is noteworthy that in 2012, in the park of Linnamäe manor in south-eastern Estonia, the entire trunk of one of Estonia’s thickest elms (CBH = 5 m) was quite massively inhabited by this species (observation by I. Süda). *S. triarmatus* has now strongly expanded its distribution in Estonia. In 2019, it was caught from Tartu, Vastseliina, Viljandi, Hummuli in southern Estonia and several sites in Tallinn, northern Estonia. In addition, all the above-mentioned vectors of DED occur in the northern European part of Russia as well, except *S. triarmatus* [75].

*Scolytus pygmaeus* has not been detected in Estonia as of yet, but recently appeared in northwest Russia in 2012 [97,100,107], being native in the central Russian territories [108]. It is likely to have spread along the roadside of the highway from Moscow to St. Petersburg [97], where planted elm stands served as corridors for leading northwards [109]; the same is true of *S. multistriatus* and *S. scolytus.* One single individual was collected from Russia, but *O. novo-ulmi* was not detected. *Scolytus pygmaeus* was found to be the vector of pathogen [110].

Ophiostomatoid fungi associate with phloem-breeding and ambrosia beetles on hardwoods [22] but there are some notes indicating higher species diversity as vectors than that previously reported from Europe [22].

*Xyleborinus saxesenii*—20% of analysed specimens contained the pathogen. The first record of *X. saxesenii* was in western Estonia from the Laulaste Nature Reserve, south-eastern Estonia in 2008 and from Matsalu, western Estonia in 2009 [111]. Later, *X. saxesenii* was found from two localities in South Estonia, Valga County: Koiva wooded meadow in 2013 and Soontaga in 2015, 2020. This work confirmed the beetle*’*s first finding in 2019 and it was caught with traps from several localities in Tallinn, northern Estonia. *Xyleborinus saxesenii* is capable of breeding in various hosts [24,93] including elms [112]. Like other ambrosia beetles, *X. saxesenii* breeds mostly in weakened or dying trees. *Xyleborinus saxesenii* has been considered an insignificant pest until recently, when it has been proved to spread *O. novo-ulmi*. Additionally, *X. saxesenii* has been shown to be able to spread the laurel wilt pathogen *Raffaelea lauricola* (Ophiostomatales) [14,112,113].

*Xyleborus dispar* and *Trypodendron signatum* are known to occur on a wide range of deciduous trees [113,114]. Both species are native, common and widespread throughout Estonia, but the latter was not found on elms. However, *Xyleborus dispar* was proved to be the vector of *O. novo-ulmi* in this work, e.g., 5% of analysed specimens contained *O. novo-ulmi*.

The implementation of such highly efficient research methods as the use of window traps has helped to detect new woodland beetle species for Estonian fauna [111,115]. On the other hand, as a result of climate change, the spread of numerous southern beetle species to the north is clearly noticeable [115], especially in the last couple of decades. The same is true for the DED vectors *Scolytus multistriatus* and *Xyleborinus saxesenii*. However, it is not clear why *Scolytus triarmatus* has become widespread and numerous in Estonia in such a short time. Considering that *Scolytus triarmatus* does not occur in Latvia, Lithuania nor Finland, the ambiguity is even greater.

### 4.3. The Traps and Alcohol as a Baiting Compound

Bark beetles are strongly attracted to synthetic pheromone [71,116] thus pheromone traps can be used to indicate their presence.

Traps used in our research were made from 1.5 L plastic bottles, because these are cost effective [92]. The number of species and specimens caught with pheromone-baited bottle traps used in this study indicate quite low efficiency as used in current work. If the pheromone was created to attract specifically for *Scolytus* spp., then ethanol is known to affect many saproxylic beetles, including bark beetles [45,117]. A comparative study in France proved that the overall specimen collecting efficiency of alcohol-baited traps in catching saproxylic beetles was twice as large as that of nonbaited traps, the efficiency ratio amounting even to 114 in catching *Xyleborinus saxesenii*! [117]. It can be assumed that in Estonia too, a relatively large number of *Xyleborinus saxesenii* and *Xyleborus dispar* specimens were lured into pheromone traps due to the attractiveness of ethanol; this does not directly indicate the high abundance of both species in the study sites. It is probable that ethanol has also an attracting effect on *Scolytus* spp. species. During the sampling, there were usually 1–2 specimens per trap at a time, so cross-contamination was diminished between same species individuals and different species. Thus, low efficiency of trapping fit well with the current work tasks.

Possible vectors could be also predators of Scolytinae: *Salpingus planirostris* and *Paromalus parallelepipedus* [118]. *Salpingus planirostris* is a common species mainly found on deciduous trees including *Ulmus*, and *P. parallelepipedus* inhabits mainly conifers, occasionally some deciduous trees as well, including *Ulmus* [49]. Therefore, it is not a coincidence that some specimens were caught on elms in Estonia.

## 5. Conclusions

This study provides new information on vectors for Dutch elm disease in northern Europe. From all 319 beetle specimens caught either with traps or handpicked—261 specimens were potential vectors of Dutch elm disease. High throughput sequencing indicated that *Ophiostoma novo-ulmi* was the most represented fungus (8.2%) on six out of seven analysed beetle species. The highest percentage of *O. novo-ulmi* was found on beetle species *S. scolytus,* followed by *X. saxesenii* and *S. triarmatus.*

According to the latest studies of potential DED vectors, the rather rapid expansion of the distribution of three species—*Scolytus triarmatus*, *S. multistriatus* and *Xyleborinus saxesenii*—in Estonia is remarkable. The first two have expanded northwards, currently spreading to the centre line of Estonia, *S. triarmatus* in addition to north-western Estonia. It is noteworthy that *S. triarmatus* and *X. saxesenii* are relatively recent newcomers in Estonia, first registered in 2004 and 2008, respectively. Whereas previously the most common *Scolytus*-species on elms in Estonia was *S. laevis*, now it seems to be *S. triarmatus*.

In conclusion, the results of our work with known elm bark beetles (*Scolytus laevis*, *S. multistriatus*, *S. triarmatus* and *S. scolytus*), indicates that there are two more species that can potentially spread Dutch elm disease in Estonia: *Xyleborus dispar* and *Xyleborinus saxesenii*. We did not catch one potential DED vector *Trypodendron signatum* on *Ulmus,* thus a connection with *O. novo-ulmi* cannot be confirmed in Estonia.

Since this was a monitoring study, there is not enough data on the extent of infection in the areas—differences between sampling sites, beetle species and sexes. A study of these named aspects could be the focus of future work.

The spread of DED should be controlled with ongoing survey of trees and the monitoring of vectoring beetles with the help of pheromone traps. Removing of all diseased elms as soon as possible is a key factor to protect healthy elms. Additionally, new and reliable species-specific DNA primers are needed for quick pathogen detection from biological samples to control disease spread.

## Figures and Tables

**Figure 1 insects-12-00393-f001:**
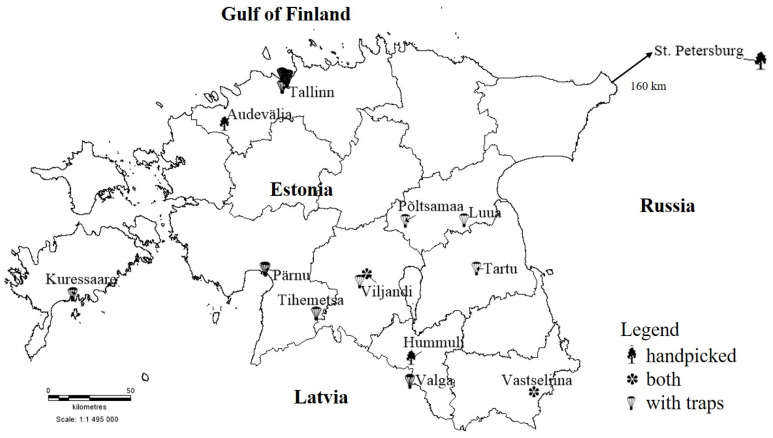
Study sites in Estonia and St. Petersburg, Russia. Bark beetles were handpicked and collected with traps or with both possibilities from the same tree.

**Figure 2 insects-12-00393-f002:**
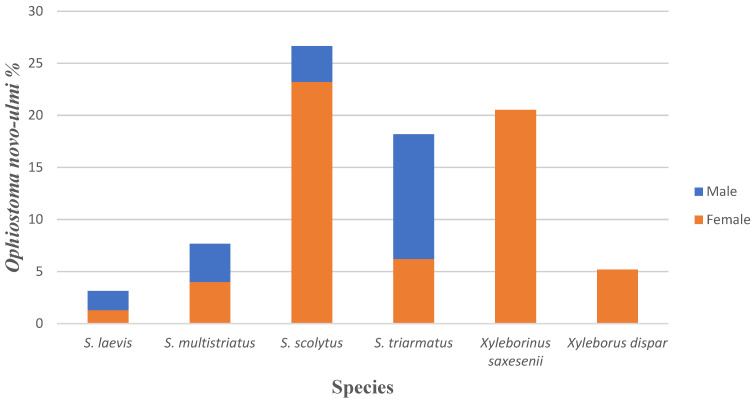
Percentage of *O. novo-ulmi* across different beetle species and gender (N = 109). No *O. novo-ulmi* was found on *S. pygmaeus.*

**Table 1 insects-12-00393-t001:** Potential vector beetle species of DED caught with traps and handpicked from symptomatic trees.

Species of Beetles	Country	Traps		Handpicked		Total
		Sex		Sex		
		Male	Female	Male	Female	
*Scolytus multistriatus*	Estonia	3	3	-	-	6
	Russia	-	-	5	-	5
*Scolytus triarmatus*	Estonia	4	2	66	114	186
*Scolytus laevis*	Estonia	-	-	21	18	39
*Scolytus scolytus*	Russia	-	-	2	2	4
*Scolytus pygmaeus*	Russia	-	-	-	1	1
*Xyleborinus saxesenii*	Estonia	-	10	-	-	10
*Xyleborus dispar*	Estonia	-	10	-	-	10
	Total	32		229		261

## Data Availability

Full ITS sequences of the entire dataset are uploaded into SRA under accession number PRJNA719602. Filtered representative full ITS sequences of *O. novo-ulmi* are uploaded into PlutoF: https://dx.doi.org/10.15156/BIO/807454, accessed on 15 April 2021.

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
