# Peer review of "Vectors of Dutch Elm Disease in Northern Europe"

_insects, 2021, doi:10.3390/insects12050393_

Round 1
Reviewer 1 Report
The authors have improved the manuscript considerably since the first version. I was particularly pleased about the complementation of Introduction and revisions of Discussion. My main concerns were all satisfactorily solved in my opinion so I recommend an acceptance of the manuscript. I have, however, some small comments that may improve the clarity:
1. Simple summary is written poorly as compared to the main text. I propose the following edits:
- line 2: last the => the last
- line 3, delete the unnecessary "It is known that" and rewrite as "Elm bark beetles Scolytus spp. are vectors of DED."
- line 3-4: "The current work's aim was" => "Our aim was"
- line 4-5: "caught with traps (please complement: bottle traps) and handpicked to identify the species (this is confusing as it sounds as if you picked the beetles from the mentioned traps by hand: you actually intend to say that you collected specimens manually) to identify the species (this is wrong, as you used the two methods to collect, not identify)."
- line 6: a whole individual => from each specimen
2. Abstract, second-last line: delete "background"
3. Other detailed comments (L=line, followed by line number in the pdf); please check the rest of the text for similar needs for small edits before submitting the final version.
L130: expired => died
L139: these are complementary data so please indicate this somehow. For example say "To secure a sufficiently large sample size for DNA analysis, we also hand-picked beetles at three sites in Estonia (how many trees?) and one site in St. Petersburg (3 trees)."
L140: the last "beetles where actively migrating on" should read "beetles were actively dispersing on" (is this really needed?). Also, it is unclear whether you collected these from bark surface or if you peeled the bark to obtain these beetles. Please complement.
L217: Within => Among
L350: delete "the species" (unnecessary words)
Reviewer 2 Report
The authors have extensively reworked the introduction and methods, and improved the presentation of results.
Please make sure that Figure 1 is high quality in the final version. At this time it is still not legible in the version I received.
Other than that I now consider the manuscript ready for publication in Insects.
